# Sphingosine 1-Phosphate (S1P) in the Peritoneal Fluid Skews M2 Macrophage and Contributes to the Development of Endometriosis

**DOI:** 10.3390/biomedicines9111519

**Published:** 2021-10-22

**Authors:** Yosuke Ono, Takako Kawakita, Osamu Yoshino, Erina Sato, Kuniyuki Kano, Mai Ohba, Toshiaki Okuno, Masami Ito, Kaori Koga, Masako Honda, Akiko Furue, Takehiro Hiraoka, Shinichiro Wada, Takeshi Iwasa, Takehiko Yokomizo, Junken Aoki, Nagamasa Maeda, Nobuya Unno, Yutaka Osuga, Shuji Hirata

**Affiliations:** 1Department of Obstetrics and Gynecology, Teine Keijinkai Hospital, Sapporo 006-08555, Japan; nadal.babolat@hotmail.co.jp (Y.O.); wa_shin_2002@yahoo.co.jp (S.W.); 2Department of Obstetrics and Gynecology, Tokushima University, Tokushima 770-8503, Japan; kawakita.takako@tokushima-u.ac.jp (T.K.); iwasa.takeshi@tokushima-u.ac.jp (T.I.); 3Department of Obstetrics and Gynecology, University of Yamanashi, Chuo 409-3898, Japan; shirata@yamanashi.ac.jp; 4Department of Obstetrics and Gynecology, Kitasato University, Sagamihara 252-0375, Japan; erinast@med.kitasato-u.ac.jp (E.S.); mhonda@kitasato-u.ac.jp (M.H.); ak.furu@kitasato-u.ac.jp (A.F.); t_hiraoka1985@yahoo.co.jp (T.H.); unno@kitasato-u.ac.jp (N.U.); 5Graduate School of Pharmaceutical Sciences, Tohoku University, Sendai 980-8578, Japan; k-kano@mol.f.u-tokyo.ac.jp (K.K.); jaoki@mol.f.u-tokyo.ac.jp (J.A.); 6Department of Biochemistry (I), Juntendo University Graduate School of Medicine, Tokyo 113-8421, Japan; mai.ohba@gmail.com (M.O.); tokuno@juntendo.ac.jp (T.O.); tyokomi@juntendo.ac.jp (T.Y.); 7Department of Obstetrics and Gynecology, University of Toyama, Toyama 930-0194, Japan; msmito@med.u-toyama.ac.jp; 8Department of Obstetrics and Gynecology, University of Tokyo, Tokyo 113-8655, Japan; kawotan-tky@umin.ac.jp (K.K.); yutakaos-tky@umin.ac.jp (Y.O.); 9Department of Obstetrics and Gynecology, Kochi University, Kochi 783-8305, Japan; maedan@kochi-u.ac.jp

**Keywords:** endometriosis, lipid, S1P, macrophage

## Abstract

Sphingosine 1-phosphate (S1P), an inflammatory mediator, is abundantly contained in red blood cells and platelets. We hypothesized that the S1P concentration in the peritoneal cavity would increase especially during the menstrual phase due to the reflux of menstrual blood, and investigated the S1P concentration in the human peritoneal fluid (PF) from 14 non-endometriosis and 19 endometriosis patients. Although the relatively small number of samples requires caution in interpreting the results, S1P concentration in the PF during the menstrual phase was predominantly increased compared to the non-menstrual phase, regardless of the presence or absence of endometriosis. During the non-menstrual phase, patients with endometriosis showed a significant increase in S1P concentration compared to controls. In vitro experiments using human intra-peritoneal macrophages (MΦ) showed that S1P stimulation biased them toward an M2MΦ-dominant condition and increased the expression of IL-6 and COX-2. An in vivo study showed that administration of S1P increased the size of the endometriotic-like lesion in a mouse model of endometriosis.

## 1. Introduction 

Endometriosis is a problematic disease in women of reproductive age, of which the primary symptoms are infertility and pelvic pain [1,2]. Inflammation is known to be involved in the onset and progression of the disease [2,3]. We also showed that aberrant secretion of inflammatory substances, such as prostaglandin, interleukin (IL)-1β, IL-6, IL-8, and monocyte chemoattractant protein (MCP)-1, play essential roles in the pathophysiology of endometriosis [4,5]. Although endometriosis is recognized as a disease of inflammation, the starting point of inflammation is still not well-understood. To understand the mechanism of inflammation in the intra-peritoneal environment, it is essential to understand the role of immune cells in the intra-peritoneal cavity [6]. Among the different types of immune cells in the intra-peritoneal cavity, it is known that the number of macrophages (MΦ) is higher in patients with endometriosis [7]. Recently, MΦ have been classified into M1 and M2, and it is known that M2 MΦ are particularly prevalent in endometriosis [8,9]. We previously showed that M2 MΦ are involved in exacerbation of endometriosis using CD206 diphtheria toxin-receptor (CD206 DTR) transgenic mice, which can deplete CD206+ M2 MΦ at any given time [10]. Therefore, understanding the starting point of inflammation and the substances that bias M2 MΦ will further clarify the pathogenesis of endometriosis.

We have been focusing on the lipid sphingosine 1-phosphate (S1P) as a mediator of inflammation [11]. S1P is a metabolite of sphingolipid, which has pleiotropic biological activities, and sphingosine kinase (SphK) catalyzes the conversion of sphingosine to S1P, a bioactive lipid [12]. S1P binds to specific G-protein-coupled receptors (S1PR 1-5) and regulates numerous cellular processes necessary for cell growth, survival, invasion, lymphocyte trafficking, vascular integrity, and production of cytokine and chemokine [12]. Therefore, the SphK–S1P–S1PR axis has attracted attention as a therapeutic target for various diseases such as cancer, rheumatoid arthritis, diabetes mellitus, and osteoporosis [12,13]. We showed that S1P is produced via SphK in endometriotic lesions and is involved in the pathogenesis of endometriosis, inducing proliferation of endometriotic cells and induction of IL-6, an inflammatory agent [11]. Moreover, Bernacchioni et al. recently reported the role of the S1P signaling axis in endometriosis-associated fibrosis [14]. S1P is also known to be abundantly contained in red blood cells and platelets [15]. Regardless of whether the theory of the etiology of endometriosis is implantation or metaplasia, it is known that the reflux of menstrual blood into the peritoneal cavity contributes to the development and progression of endometriosis [2]. Therefore, we hypothesized that the S1P concentration in the peritoneal cavity would increase during menstruation due to the reflux of menstrual blood, and investigated the S1P concentration in human peritoneal fluid (PF). In addition, we investigated the relationship between S1P and MΦ, especially M2 MΦ.

## 2. Materials and Methods

### 2.1. Reagents and Materials

Roswell Park Memorial Institute (RPMI)-1640 medium and fetal bovine serum (FBS) was obtained from Life Technologies (Tokyo, Japan). Antibiotics (a mixture of penicillin, streptomycin, and amphotericin B) were obtained from Wako Pure Chemical Industries (Osaka, Japan). S1P was purchased from Sigma (St. Louis, MO, USA). VPC 23019, an antagonist of S1PR-1 and S1PR-3, was purchased from Tocris Inc. (Ellisville, MO, USA). The experimental procedures were approved by the institutional review board of the University of Toyama, Kitasato University and Tokushima University. Signed informed consent for the use of samples was obtained from each patient. Human samples, including peritoneal cells, were obtained from patients undergoing laparoscopic surgery for benign gynecological diseases. All women had regular menstrual cycles, and none had been taking hormonal medications for at least 3 months prior to laparoscopy.

### 2.2. Collection of Peritoneal Fluid (PF) and Culture of MΦ Derived from the Peritoneal Fluid Mononuclear Cell (PFMC)

Women of reproductive age who underwent laparoscopic surgery for benign diseases were included in this study. All subjects in this study were Japanese, and the mean ages of the non-endometriosis group (*N* = 19) and the endometriosis group (*N* = 14) were 34.2 ± 6.4 (mean ± SD) and 35.6 ± 6.0 years, respectively, with no differences between the two groups. All endometriosis patients had endometriomas and revised American Society of Reproductive Medicine (re-ASRM) stage III or IV endometriosis. The PF was aspirated from the pouch of Douglas immediately after insertion of the trocar to minimize contamination with blood. To obtain PFMsC, the PFs were treated with Lymphoprep (Axis-Shield PoC AS, Oslo, Norway) [16]. Briefly, the PFs were layered onto Lymphoprep, centrifuged at 377× *g* for 30 min, and PF cells were collected from the interface and washed with PBS. CD14 Microbeads (Miltenyi Biotec, Bergisch Gladbach, Germany) were used for separating MΦ from isolated PF cells. The separated MΦ were resuspended in RPMI-1640 medium containing 10% fetal bovine serum and plated at a density of 2 × 10^5^ cells/mL in 12-well culture plates. After overnight incubation, these cells were cultured in replenished serum-free media without (control) or with S1P (125 nM) for another 8 h to examine mRNA levels, and for another 24 h to examine protein levels. In a pilot study using various concentrations of S1P, we confirmed that stable results were obtained with S1P of 125 nM [11].

### 2.3. Mass Spectrometry

The concentration of S1P and total PG-E2 and its metabolites, 14-dihydro-15-keto-*PGE2* (dhk PG-E2), in PF were measured using mass spectrometry. The S1P level was measured in 19 non-endometriosis and 14 endometriosis patients. PG-E2 and dhk PG-E2 were measured in 5 non-endometriosis patients (age; 32.1 ± 7.2 (mean ± SD)) and 9 endometriosis patients (age; 35.8 ± 7.0). There was no difference in age between the two groups. Samples were measured in triplicate, and mean values were obtained.

### 2.4. In Vivo Study Using a Mouse Model of Endometriosis 

This study was approved by the Committee of the Institute of Animal Experimental of Tokushima Graduate School (T2020-123). Eight-week-old female C57/6J mice were purchased from CLEA Japan, Inc. (Tokyo, Japan). The mice were fed on a mouse diet and water ad libitum and kept in a light/dark cycle of 12/12 h under controlled conditions. Before any invasive procedure, the mice were anesthetized with sevoflurane. The surgical technique was performed under sterile conditions. The continuous administration of estrogen was carried out through a subcutaneously implanted tube [17]. Donor mice were killed 14 days after estrogen implantation. Blood was drawn from the inferior vena cava, and the uterus was collected. Endometrial fragments derived from one side of the uterine horn were suspended in 0.1 mL of sterilized saline and 0.2 mL of blood. A 2–3 mm incision was performed in the left lower abdomen of recipient mice, and the endometrial fragments of the donor mouse were administered intraperitoneally. On the same day, S1P (400 ng/mouse) or vehicle (PBS) was administered for 5 consecutive days. On day 6, the recipient mice were euthanized and the peritoneal cavity of each mouse was examined. The total area (mm^2^) of endometriotic-like lesions was measured for each mouse. The lesions were removed, fixed in 4% paraformaldehyde, and embedded in paraffin.

### 2.5. Immunohistochemistry Study

Paraffin-embedded tissues were cut into 5 µm thick sections and mounted on slides. Tissues were deparaffinized with xylene, rehydrated through a graded series of ethanol, and washed with water. Antigen retrieval was performed in 10 mM sodium citrate buffer (pH 6.0) by heating in a microwave for 10 min and then cooling to room temperature. Rabbit IgG was used as a negative control. Slide staining with the first and second antibodies was performed according to the manufacturer’s instructions [18]. Immunofluorescence analysis of endometriotic-like lesions was performed using rabbit anti-mannose receptor (CD206) (Abcam, Cambridge, UK Cat# 64693, 1:100 dilution). The primary antibody was incubated overnight at 4 °C. As the secondary antibody, rat anti-rabbit antibody was used. We used 4′,6-diamidino-2-phenylindole (DAPI; 1:500) to detect nuclei. Rabbit normal IgG was used instead of the primary antibody as the negative control. All images were captured using a Keyence BZ-X800 (Keyence, Tokyo, Japan).

### 2.6. Reverse Transcription (RT) and Quantitative Real-Time Polymerase Chain Reaction (PCR) Analysis 

Total RNA was extracted from human PF MΦ using a ISOGEN-II (NIPPON GENE, Tokyo, Japan). RT was performed using Rever Tra Ace qPCR RT Master Mix with gDNA Remover (TOYOBO, Tokyo, Japan). Approximately 0.5–1 μg of total RNA was reverse-transcribed in a 20 μL volume. For the quantification of various mRNA levels, real-time PCR was performed using a Mx3000P Real-Time PCR System (Agilent Technologies, Santa Clara, CA, USA) according to the manufacturer’s instructions. The PCR primers used with SYBR Green methods were selected from different exons of the corresponding genes to discriminate PCR products that might arise from possible chromosomal DNA contaminants. The SYBR Green thermal cycling conditions were 1 cycle of 95 °C for 30 s, and 40 cycles of 95 °C for 10 s, 60 °C for 10 s, and 72 °C for 10 s. The relative mRNA levels were calculated using the standard curve method and were normalized to the mRNA levels of GAPDH (forward: AATGTGTCCGTCGTGGATCTGA and reverse: GATGCCTGCTTCACCACCTTCT). The relative mRNA levels were calculated using the standard curve method and were normalized to the mRNA levels of GAPDH. Primers sequences of CD163, IL-6, and COX-2 are: CD163 (forward, GAACATGTCACGCCAGC and reverse, CGAGTTAACGCCAGTAAGG), IL-6 (forward, ACAAGCCAGAGCTGTGCAGATG and reverse, GTGCCCATGCTACATTTGCCGA), and COX-2 (forward, AGATCATCTCTGCCTGAGTATCTT and reverse, TTCAAATGAGATTGTGGGAAAATTGC). 

### 2.7. Cytokine Measurement

Concentrations of IL-6 in cell culture supernatants were measured using enzyme-linked immunosorbent assay (ELISA Kit (Cloud-Clone, Katy, TX, USA)). The limit of sensitivity was 0.1 pg/mL. The intra- and inter-assay coefficients of variation were less than 5% in the assays.

### 2.8. Statistical Analysis

Data were evaluated by the Mann–Whitney U-test. A *p*-value less than 0.05 was accepted as statistically significant.

## 3. Results

### 3.1. The Concentrations of S1P, Total PG-E2, and Its Metabolites (dhk PG-E2) in PF

The concentration of S1P in the PF was measured using mass spectrometry. The concentration of S1P during menstrual period (*N* = 9; mean ± SEM: 97.7 ± 21.3 nM; median: 74.0 nM) was significantly higher than that during non-menstrual period (*N* = 24; 42.6 ± 7.1 nM; median 36.1 nM, *p* < 0.05; Figure 1a). When samples from all periods of the menstrual cycles were combined, there was no difference in S1P concentration between the non-endometriosis (*N* = 19; 53.4 ± 13.0 nM; median 36.3 nM) and endometriosis (*N* = 14; 63.5 ± 8.7 nM; median 63.5 nM) patients (Figure 1b). Focusing on a specific time of menstrual cycle, during the non-menstrual period, endometriosis patients showed a significant increase in S1P concentration (*N* = 10; 58.5 ± 10.0 nM; median 52.0 nM) compared to that of the non-endometriosis patients (*N* = 14; 31.4 ± 4.6 nM; mean 29.4 nM; *p* < 0.05; Figure 1c). In contrast, there was no difference in S1P concentration between the endometriosis patients (*N* = 4; 76.0 ± 11.8 nM; median 70.6 nM) and non-endometriosis patients (*N* = 5; 118 ± 37.2 nM; median 108 nM) during the menstrual period (Figure 1d). In addition, in the endometriosis patients, the total concentration of prostaglandin-E2 and its metabolites (dhk PG-E2) was significantly increased (*N* = 9; 43.3 ± 15.6 nM; median 28.2 nM) than that of non-endometriosis group (*N* = 5; 8.4 ± 4.2 nM; median 3.4 nM; *p* < 0.05; Figure 1e).

### 3.2. The Effect of S1P on Human PF MΦ

As shown in Figure 2, S1P stimulation (125 nM) of PF MΦ significantly increased the mRNA expression of CD163, a human M2 MΦ marker, by an average of 2.6-fold, suggesting the induction to M2 MΦ in peritoneal environment (*p* < 0.05, Figure 2a). Furthermore, S1P significantly increased the mRNA expressions of IL-6 by an average of 2.6-fold (*p* < 0.05) and COX-2 by an average of 14.4-fold (*p* < 0.01) (Figure 2b,c). We also found that twenty-four hours after S1P stimulation, the IL-6 protein level in the culture supernatant was 1.3-fold higher compared to the control (*p* < 0.05), which was canceled by VPC 23019 (5 μM), an antagonist of S1PR-1 and 3 (Figure 2d). 

### 3.3. The Effect of S1P on the Endometriotic-Like Lesion in an Endometriosis Mouse Model

We examined the effect of S1P on endometriosis lesions in a mouse model and found that the area of endometriotic-like lesions of S1P-treated mice was significantly larger compared to the control group (mean ± SEM; 1.3 ± 0.8 vs. 17.9 ± 6.8 mm^2^, median; 1.0 vs. 5.5 mm^2^, *p* < 0.05; Figure 3a,b), but there was no difference in the number of endometriotic-like lesions (data not shown). Our immunofluorescence study revealed that the expression of CD206+M2MΦ was observed in endometriotic-like lesions, especially localized in stromal cells, in S1P-treated mice (Figure 4). The negative control (rabbit IgG) did not show any immunostaining (data not shown).

## 4. Discussion

An epidemiological study of endometriosis showed that increased frequency of menstruation and increased amount of refluxed menstrual blood increase the risk of developing endometriosis [19]. It is expected that the refluxed menstrual blood contains blood components other than endometrium. Therefore, we focused on S1P, which is abundant in blood components, especially in red blood cells and platelets [15]. The relationship between red blood cells and endometriosis was reported: excess iron accumulation can result in toxicity and may contribute to the development of endometriosis [20]. There have been some studies on platelets and endometriosis. Zhang et al. reported that platelets promote endometriotic lesions by inducing angiogenesis and facilitating epithelial–mesenchymal transition and fibroblast-to-myofibroblast transdifferentiation through activation of the TGF-β/Smad3 signaling pathway [21,22].

In the present study, we found that S1P levels were predominantly elevated during the menstrual period compared to the non-menstrual period, regardless of the presence or absence of endometriosis. This suggests that at least some of the S1P may be derived from refluxing blood. Furthermore, patients with endometriosis had significantly elevated S1P levels compared to the control during the non-menstrual period. 

In the previous study, we confirmed that S1P induces the proliferation of endometriotic cells [11]. Our group also observed that intraperitoneal administration of blood contributes to disease progression in endometriosis model [17]. One of the mechanisms by which blood may be an aggravating factor of endometriosis is the increased concentration of S1P in the peritoneal environment. In the present study, when S1P was administered to a mouse model of endometriosis, an increase in lesion size with CD206+ M2 MΦ was observed. We also found that treatment of human MΦ with S1P induced CD163+ M2 MΦ. In the peritoneal cavity of patients with endometriosis, MΦ are thought to be biased to M2 MΦ [8]. Previously, we showed that M2 MΦ contribute to angiogenesis through the production of TGF-β, and that this leads to exacerbation of endometriosis [10]. However, what causes the bias toward M2 MΦ in endometriosis patients is not well-understood. In this study, we found that S1P increased the M2 MΦ marker, suggesting that chronically high levels of S1P may contribute to the M2 bias of MΦ in endometriosis patients.

S1P also induced the expression of the pro-inflammatory cytokines IL-6 and COX-2 in MΦ. In this study, we measured PG-E2, a product of COX-2, and its metabolites in the PF and found that they were clearly increased in endometriosis patients compared to non-endometriosis patients. Cheng et al. reported that PG-E2 reduces the phagocytic ability of peritoneal MΦ in endometriosis patients [23]. One can speculate that the excessive presence of COX-2 expressing MΦ in endometriosis patients may inhibit the removal of refluxed menstrual blood, contributing to high S1P levels during the non-menstrual phase. Collectively, it is suggested that MΦ in patients with endometriosis may be trapped in a vicious cycle of decreased phagocytosis of MΦ via increased COX-2 expression due to S1P stimulation, and further increase of S1P in the peritoneal cavity due to decreased phagocytosis (Figure 5).

Considering the application of our findings to the treatment of endometriosis, breaking up of the vicious cycle of increased COX-2 expressing M2 MΦ and S1P could be a potential therapeutic target. There are three possible strategies to tackle this issue: (1) Reducing retrograde endometrium: low-dose estrogen-progestin (LEP) may be effective. Whether the use of LEP decreases the intraperitoneal S1P concentration needs to be investigated. (2) Reducing S1P levels: recently, new therapeutic approaches using S1P-neutralizing antibody [24] or S1P receptors antagonists [25,26] have been developed and are being tested in clinical trials for cancer and age-related macular degeneration. (3) Modulation of MΦ: Theoretically, suppression of COX-2 could be a treatment for endometriotic lesions. Efstathiou et al. reported that chronic administration of non-steroidal anti-inflammatory drugs (NSAIDs) limits the progression of endometriosis in a murine model [27]. Additionally, since there is plasticity between M1 and M2 MΦ, shifting to M1 MΦ using GM-CSF or palmitate [28,29] may reduce the activity of endometriosis, which warrants examination.

The limitation of this study is that the sample size of human clinical specimens was relatively small. Further accumulation of human samples, including hormone treatment cases, is needed.

## Figures and Tables

**Figure 1 biomedicines-09-01519-f001:**
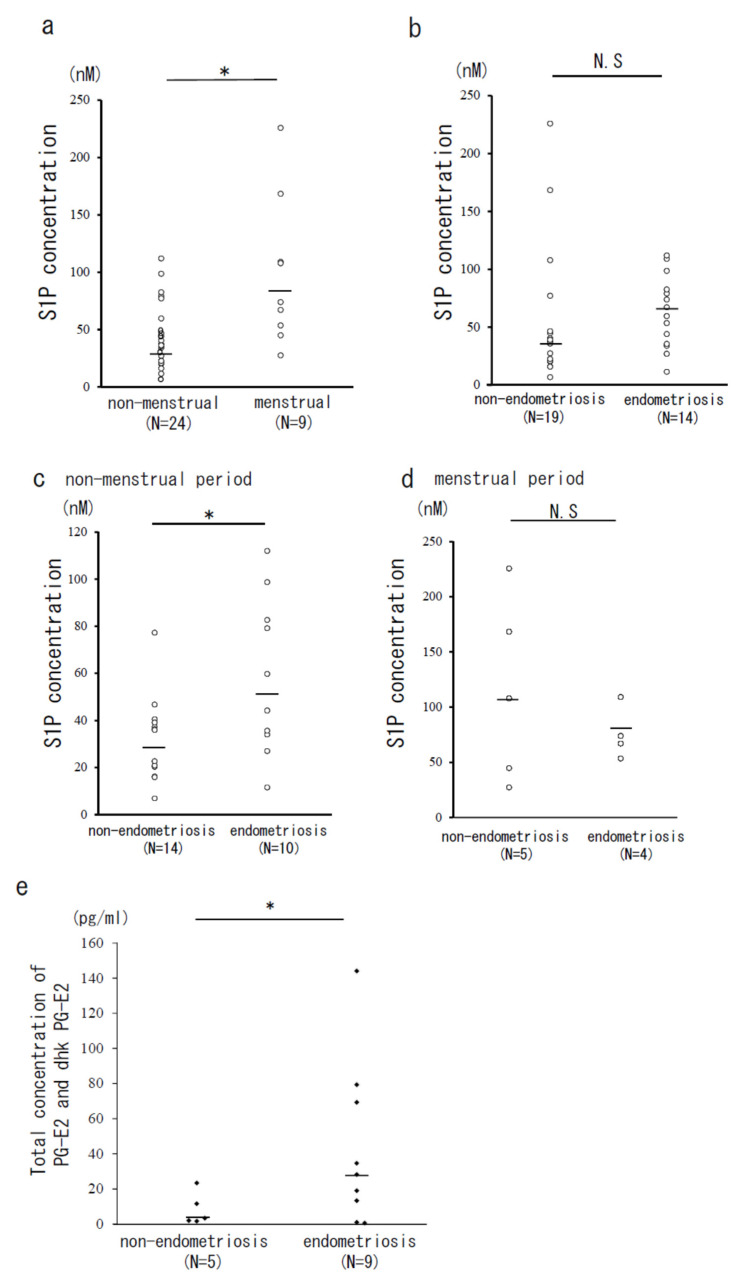
The concentration of S1P and prostaglandin (PG)-E2 with its metabolites (dhk PG-E2) in peritoneal fluid measured by mass spectrometry. The peritoneal S1P levels were compared between the patients during the non-menstrual period (*N* = 24) and the menstrual period (*N* = 9) (**a**), and between the non-endometriosis (*N* = 19) and endometriosis patients (*N* = 14) (**b**). The S1P levels in the non-menstrual (*N* = 14 vs. *N* = 10, (**c**) and the menstrual periods (*N* = 5 vs. *N* = 4, (**d**) of non-endometriosis and endometriosis patients were compared. The concentration of PG-E2 and its metabolites (dhk PG-E2) was compared between the non-endometriosis (*N* = 5) and endometriosis patients (*N* = 9) (**e**). Bars show the median value. *, *p* < 0.05; N.S., not significant.

**Figure 2 biomedicines-09-01519-f002:**
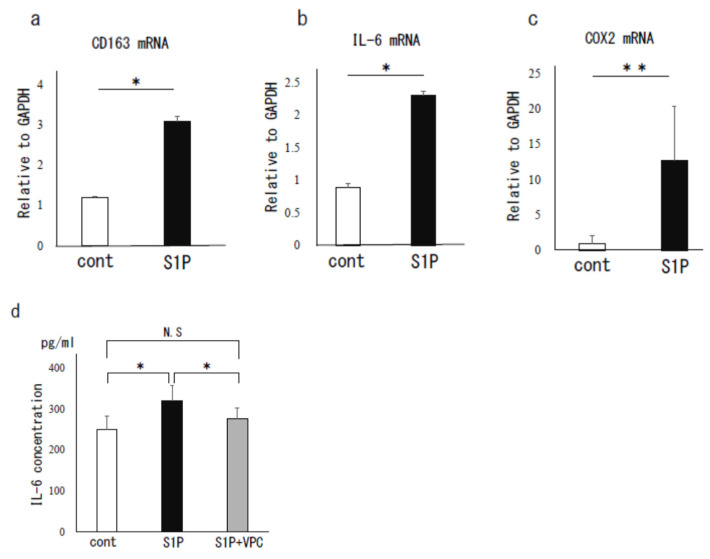
The S1P effect on human peritoneal macrophages (MΦ). Peritoneal MΦ were stimulated with S1P (125 nM) for 8 h. Total RNA was extracted from the cells and subjected to real-time PCR to determine the mRNA levels of CD163 (**a**), IL-6 (**b**), and COX-2 (**c**). Data were normalized by GAPDH mRNA levels to show the relative abundance. The supernatant of peritoneal MΦ stimulated with S1P (125 nM, 24 h) was subjected for measurement of IL-6 protein by ELISA. VPC23019, an antagonist of S1P receptor (S1PR)-1 and S1PR-3, was used to antagonize the S1P effect (**d**). Data from 3 different experiments are shown as the mean ± SEM. *, *p* < 0.05 (vs. control); **, *p* < 0.01 (vs. control); N.S., not significant, cont., control.

**Figure 3 biomedicines-09-01519-f003:**
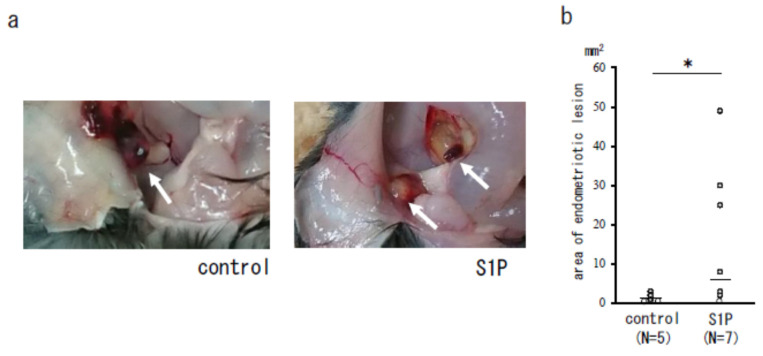
The S1P effect on an endometriosis mouse model. S1P (400 ng/mouse) or control (PBS) was administered for five consecutive days starting from the day the endometrium was inoculated into the peritoneal cavity of the mice. The recipient mice were euthanized on day 6, and the peritoneal cavity of each mouse was examined. The white arrows indicate the endometriotic-like lesions (**a**). Endometriotic-like lesions were measured for the total area of the lesion (mm^2^) per mouse (**b**). Bars show the median value. *, *p* < 0.05.

**Figure 4 biomedicines-09-01519-f004:**
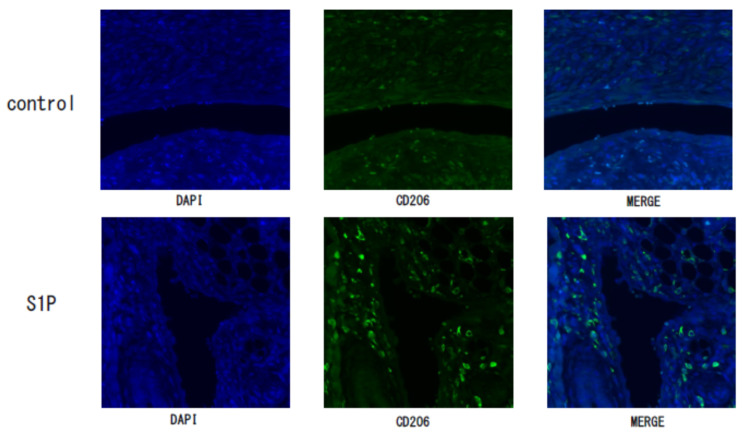
The expressions of CD206 in the endometriotic-like lesion in control and S1P-treated endometriosis mouse model. Endometriotic-like lesions from control or S1P-treated mice were stained with an antibody of CD206, a marker of macrophages. As a secondary antibody, the gout anti-rabbit antibody was used and 4′,6-diamidino-2-phenylindole (DAPI) was used to detect nuclei.

**Figure 5 biomedicines-09-01519-f005:**
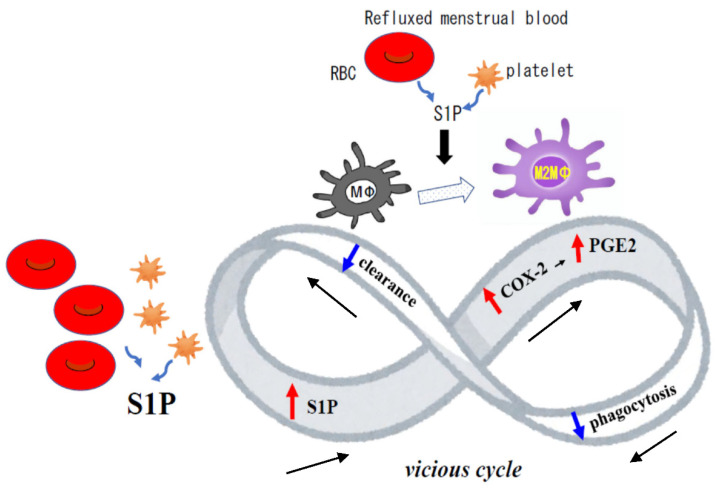
Schema of the hypothesis: a vicious cycle. S1P, which is rich in red blood cells (RBC) and platelets, skewed M2 macrophages (MΦ), and induced COX-2 expression in the peritoneal MΦ. Decreased phagocytosis of MΦ caused by increased prostaglandin (PG)-E2 [23] may suppress its effect of clearance of peritoneal cavity, resulting in a higher level of S1P concentration in the peritoneal cavity. Further increase in S1P in the peritoneal cavity may induce more COX-2 expression, followed by a further decrease in phagocytosis of MΦ.

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
