# Peer review of "Sphingosine 1-Phosphate (S1P) in the Peritoneal Fluid Skews M2 Macrophage and Contributes to the Development of Endometriosis"

_biomedicines, 2021, doi:10.3390/biomedicines9111519_

Round 1

Reviewer 1 Report

Ono et al. have investigated the S1P levels in the human peritoneal fluid during menstrual and non-menstrual phases in patients with endometriosis and compared them with controls. During the non-menstrual phase, they have found an increase in S1P in patients with endometriosis compared to controls. They have performed experiments with human peritoneal macrophages incubated with S1P, observing and increase in IL-6 and COX-2 expression, as well as in vivo studies and they have found that S1P increases of the endometriotic-like lesions in a mice model. The experiments are well performed and both the introduction and the discussion are clear. Although the reader will want to know whether decreasing S1P levels or using S1P receptor antagonists impact endometriotic lesions, I recognized that this could be the aim of a different paper and the results described in this manuscript are suitable for publication.

I have only a minor point in line  117,  I do not understand what the Authors mean with recombinant S1P. I guess that this is just a typo.

Recently, Bernacchioni et al. (doi: 10.1016/j.fertnstert.2020.08.012) have reported that S1P receptors are dysregulated in endometriosis. They Authors could consider to include this information in the discussion.

Ono et al. have investigated the S1P levels in the human peritoneal fluid during menstrual and non-menstrual phases in patients with endometriosis and compared them with controls. During the non-menstrual phase, they have found an increase in S1P in patients with endometriosis compared to controls. They have performed experiments with human peritoneal macrophages incubated with S1P, observing and increase in IL-6 and COX-2 expression, as well as in vivo studies and they have found that S1P increases of the endometriotic-like lesions in a mice model. The experiments are well performed and both the introduction and the discussion are clear. Although the reader will want to know whether decreasing S1P levels or using S1P receptor antagonists impact endometriotic lesions, I recognized that this could be the aim of a different paper and the results described in this manuscript are suitable for publication.

I have only a minor point in line  117,  I do not understand what the Authors mean with recombinant S1P. I guess that this is just a typo.

Recently, Bernacchioni et al. (doi: 10.1016/j.fertnstert.2020.08.012) have reported that S1P receptors are dysregulated in endometriosis. They Authors could consider to include this information in the discussion.

Author Response

Ono et al. have investigated the S1P levels in the human peritoneal fluid during menstrual and non-menstrual phases in patients with endometriosis and compared them with controls. During the non-menstrual phase, they have found an increase in S1P in patients with endometriosis compared to controls. They have performed experiments with human peritoneal macrophages incubated with S1P, observing and increase in IL-6 and COX-2 expression, as well as in vivo studies and they have found that S1P increases of the endometriotic-like lesions in a mice model. The experiments are well performed and both the introduction and the discussion are clear. Although the reader will want to know whether decreasing S1P levels or using S1P receptor antagonists impact endometriotic lesions, I recognized that this could be the aim of a different paper and the results described in this manuscript are suitable for publication.

>Thank you very much for your positive comments. As you pointed out, we will continue to investigate the use of antagonists of S1P receptor as an extension of this paper. Thank you very much for your fruitful advice.

I have only a minor point in line  117,  I do not understand what the Authors mean with recombinant S1P. I guess that this is just a typo.

>Thank you very much. We have removed the word "recombinant" as it was confusing.

Recently, Bernacchioni et al. (doi: 10.1016/j.fertnstert.2020.08.012) have reported that S1P receptors are dysregulated in endometriosis. They Authors could consider to include this information in the discussion.

>We appreciate your suggestion. In the revised version, we have added the paper pointed out.

(page 2, line 60-61)

Moreover, Bernacchioni et al. recently reported the role of the S1P signaling axis in endometriosis-associated fibrosis [14].

Reviewer 2 Report

Journal: BioMed

Manuscript ID: biomedicines-1404372

Title: Sphingosine 1-phosphate (S1P) in the peritoneal fluid skews M2 macrophage and contributes to the development of endometriosis

The authors hypothesized that the concentration of S1P in the peritoneal cavity may increase during menstruation due to reflux of menstrual blood, and investigated the concentration of S1P in humans peritoneal fluid. Furthermore, the relationship between S1P and macrophages, especially M2 macrophages, was examined in this study.

Comments and Suggestions for Authors:

The manuscript is an interesting study, but requires some considerations.

The manuscript presents two related but different studies and could be published separately.

  1. Introduction:

Page 1, Line 42. There is a space on this line of the manuscript that should be removed.

  1. Material and methods:

It remains to describe the characteristics of the patients in the study (demographic, clinical, social).

Page 2, Lines 79 to 81. This paragraph is repeated on lines 105 to 106.

  1. Results:

Firstly, the homogeneity of the biological variables of the groups of patients considered should be assessed.

Page 4, Line 184. There is a space on this line of the manuscript that should be removed.

  1. Discussion:

A Discussion section in which the authors compare their findings with other authors is missing. In general, little is disputed with the results of other similar studies.

The authors do not acknowledge the limitations of the study, taking into account sources of potential bias or imprecision, such as the small number of cases considered. Has the sample size been calculated previously?

  1. Conclusions

This section is left long for Conclusions and in part should move on to the Discussion section.

References:

References should be thoroughly revised to conform to uniform and appropriate standards for the journal BioMed.

Author Response

・The authors hypothesized that the concentration of S1P in the peritoneal cavity may increase during menstruation due to reflux of menstrual blood, and investigated the concentration of S1P in humans peritoneal fluid. Furthermore, the relationship between S1P and macrophages, especially M2 macrophages, was examined in this study.

Comments and Suggestions for Authors:

・The manuscript is an interesting study, but requires some considerations.

The manuscript presents two related but different studies and could be published separately.

>We appreciate your favorable comment. We also considered splitting the paper into two, but since this would weaken the content of each paper, we decided to submit them together this time. Thank you very much for your insightful suggestion.

Introduction:

・Page 1, Line 42. There is a space on this line of the manuscript that should be removed.

>Thank you for pointing this out. The line has been changed.

Material and methods: 

・It remains to describe the characteristics of the patients in the study (demographic, clinical, social).

>Thank you for pointing this out. For better understanding, we have changed the order of the descriptions in the material and method section. We looked at the data and found that all of the samples were Japanese women. We also looked at the age of the women and found no difference in age between the non-endometriosis group and the endometriosis group. We added the sentences below in the revised manuscript.

(page 2, line 86-88)

All subjects in this study were Japanese, and the mean ages of the non-endometriosis group (N=19) and the endometriosis group were 34.2 ± 6.4 (mean ± SD) and 35.6 ± 6.0 years, respectively, with no difference between the two groups.

(page 3, line 107-109)

PG-E2 and dhk PG-E2 was measured in 5 endometriosis patients (age; 32.1± 7.2 (mean ± SD)) and 9 non-endometriosis patients (age; 35.8± 7.0). There was no difference in age between the two groups.

・Page 2, Lines 79 to 81. This paragraph is repeated on lines 105 to 106.

>We apologize for the repetition of the same message. The latter sentence has been deleted. Thank you very much for pointing it out.

Results:

・Firstly, the homogeneity of the biological variables of the groups of patients considered should be assessed.

>Thank you for your suggestion. First of all, we compared the age groups of the non-endometriosis group and the endometriosis group and showed that there was no difference. In addition, we had stated that we did not use hormone preparations in all cases. In the material and method section, it was noted that the endometriosis cases were in stages III and IV of the re-ASRM classification. In the revised version, we added the sentences below.

(page 2, line 86-90)

All subjects in this study were Japanese, and the mean ages of the non-endometriosis group (N=19) and the endometriosis group were 34.2 ± 6.4 (mean ± SD) and 35.6 ± 6.0 years, respectively, with no difference between the two groups. All endometriosis patients had endometriomas and revised American Society of Reproductive Medicine (re-ASRM) stage III or IV endometriosis.

(page 3, line 107-109)

PG-E2 and dhk PG-E2 was measured in 5 endometriosis patients (age; 32.1± 7.2 (mean ± SD)) and 9 non-endometriosis patients (age; 35.8± 7.0). There was no difference in age between the two groups.

・Page 4, Line 184. There is a space on this line of the manuscript that should be removed.

>Thank you for pointing this out. The line has been changed.

Discussion:

・A Discussion section in which the authors compare their findings with other authors is missing. In general, little is disputed with the results of other similar studies.

>Thank you for your suggestion. In the revised version, we have added the previous papers on red blood cells, platelets and endometriosis in particular. We added the sentences below in the revised version.

(page 8, line 250-256)

The relationship between red blood cells and endometriosis has been reported that excess iron accumulation can result in toxicity and may be contributing to the development of endometriosis[21]. There have been some studies on platelets and endometriosis. Zhang et al. reported that platelets promote the endometriotic lesions by inducing angiogenesis and facilitating epithelial-mesenchymal transition and fibroblast-to-myofibroblast transdifferentiation through activation of the TGF-β/Smad3 signaling pathway[22][23]

・The authors do not acknowledge the limitations of the study, taking into account sources of potential bias or imprecision, such as the small number of cases considered. Has the sample size been calculated previously?

>Thank you for pointing this out. As you pointed out, the limitation of the study is that the number of clinical samples is small. In addition, we did not consider the sample size beforehand and conducted the experiment. We think that the accumulation of cases is necessary in the future. We have added the following sentence in the text.

(page 9, line300-302)

The limitation of this paper is that the sample size of human clinical specimens is relatively small. Further accumulation of human samples, including hormone treatment cases, is needed.

Conclusions

・This section is left long for Conclusions and in part should move on to the Discussion section.

>We are very sorry, all the sentences listed in the” Couclusions “ should be listed in the discussion part. In Revise, we will change it to 4. Discusssion section.

References:

・References should be thoroughly revised to conform to uniform and appropriate standards for the journal BioMed.

>Thank you for pointing this out. We have formatted the referrals in the revised version.

Round 2

Reviewer 2 Report

Journal: BioMed

Manuscript ID: biomedicines-1404372

Title: Sphingosine 1-phosphate (S1P) in the peritoneal fluid skews M2 macrophage and contributes to the development of endometriosis

In the new V2 manuscript, the authors have made changes based on the recommendation of referees that improve its presentation. The authors honestly acknowledge the main limitation in the last paragraph of the manuscript, due to the small number of cases they contribute to support their results. However, this limitation should be extended to the Abstract section.

Author Response

The authors honestly acknowledge the main limitation in the last paragraph of the manuscript, due to the small number of cases they contribute to support their results. However, this limitation should be extended to the Abstract section.

>Thank you very much for your advice. As advised, in the revised version, we have added the number of patients, and the description that "the relatively small number of samples requires caution in interpreting the results".

Now it reads as below (abstract part)  The text in red is the corrected part.

We hypothesized that the S1P concentration in the peritoneal cavity would increase especially during the menstrual phase, due to the reflux of menstrual blood, and investigated the S1P concentration in the human peritoneal fluid (PF) from 14 non-endometriosis and 19 endometriosis patients. Although the relatively small number of samples requires caution in interpreting the results, S1P concentration in the PF during the menstrual phase was predominantly increased compared to the non-menstrual phase, regardless of the presence or absence of endometriosis.